# Highly Sensitive and Rapid Characterization of the Development of Synchronized Blood Stage Malaria Parasites Via Magneto-Optical Hemozoin Quantification

**DOI:** 10.3390/biom9100579

**Published:** 2019-10-07

**Authors:** Mária Pukáncsik, Petra Molnár, Ágnes Orbán, Ádám Butykai, Lívia Marton, István Kézsmárki, Beáta G. Vértessy, Mohd Kamil, Amanah Abraham, Ahmed S. I. Aly

**Affiliations:** 1Department of Physics, Budapest University of Technology and Economics and MTA-BME Lendület Magneto-optical Spectroscopy Research Group, 1111 Budapest, Hungary; orbanag@gmail.com (Á.O.); butykai.adam@gmail.com (Á.B.); kezsmark@mail.bme.hu (I.K.); 2Institute of Enzymology, Research Centre for Natural Sciences, Hungarian Academy of Sciences, 1117 Budapest, Hungary; marton.livia@ttk.mta.hu (L.M.); vertessy@mail.bme.hu (B.G.V.); 3Department of Applied Biotechnology, Budapest University of Technology and Economics, 1111 Budapest, Hungary; 4Beykoz Institute of Life Sciences and Biotechnology, Bezmialem Vakif University, Istanbul 34093, Turkey; mkamil@bezmialem.edu.tr; 5Department of Tropical Medicine, Tulane University, New Orleans, LA 70112, USA; aabraham@tulane.edu

**Keywords:** malaria, *Plasmodium vinckei*, *Plasmodium yoelii*, rotating-crystal magneto-optical diagnostic, hemozoin, synchronous blood stage

## Abstract

The rotating-crystal magneto-optical diagnostic (RMOD) technique was developed as a sensitive and rapid platform for malaria diagnosis. Herein, we report a detailed in vivo assessment of the synchronized *Plasmodium vinckei* lentum strain blood-stage infections by the RMOD method and comparing the results to the unsynchronized *Plasmodium yoelii* 17X-NL (non-lethal) infections. Furthermore, we assess the hemozoin production and clearance dynamics in chloroquine-treated compared to untreated self-resolving infections by RMOD. The findings of the study suggest that the RMOD signal is directly proportional to the hemozoin content and closely follows the actual parasitemia level. The lack of long-term accumulation of hemozoin in peripheral blood implies a dynamic equilibrium between the hemozoin production rate of the parasites and the immune system’s clearing mechanism. Using parasites with synchronous blood stage cycle, which resemble human malaria parasite infections with *Plasmodium falciparum* and *Plasmodium vivax*, we are demonstrating that the RMOD detects both hemozoin production and clearance rates with high sensitivity and temporal resolution. Thus, RMOD technique offers a quantitative tool to follow the maturation of the malaria parasites even on sub-cycle timescales.

## 1. Introduction

The malaria parasite exhibits a complex life cycle with multiple stages developing in and migrating through various tissues inside the mosquito vector and the mammalian host. However, the disease manifestations and burden of malaria is solely caused by the asexual blood stage parasites invading and developing inside erythrocytes. Therefore, the blood stage parasites have always been the main focus of diagnostics, vaccine and drug development.

In endemic areas the diagnosis of malaria is still widely carried out by the analysis of Giemsa-stained blood smears using light microscopy due to the relative affordability and reliability of this method. However, diagnosis of malaria by light microscopy is often not sensitive enough to detect very low parasite numbers, extensively time-consuming and requires well-trained personnel [1]. Indeed, a highly sensitive, instant, reliable and cost effective method for malaria diagnosis is still urgently needed.

The highly sensitive and cost-effective rotating crystal magneto-optical diagnostic (RMOD) technique, potentially fulfilling the aforementioned criteria, has been developed for clinical and in-field diagnosis of malaria [2,3,4]. The method quantifies hemozoin in lysed blood samples by exploiting its unique magnetic and optical anisotropy [3]. This unique heme compound is produced by all the species of *Plasmodium* as a by-product of hemoglobin digestion during the parasites’ intra-erythrocytic developmental cycle, and its concentration was shown to be directly correlated with parasitemia [5,6]. The preliminary validation of the RMOD technique as a diagnostic tool was performed on *Plasmodium falciparum* in vitro cultures and on mice infected with *Plasmodium berghei* ANKA parasites, demonstrating its potential for field-based diagnostics [3,4].

In humans and other mammalian hosts, the parasites usually exhibit a synchronous intra-erythrocytic blood stage cycle, in which the simultaneous egress of merozoites from infected red blood cells results in the release of a significant amount of hemozoin crystals into the blood steam. The simultaneous release of hemozoin, erythrocyte residues and parasite-derived toxins triggers malaria pathology and elicit elements of the immune system [7,8,9].

The released hemozoin gets rapidly phagocytosed by monocytes and macrophages and it can persist within them for long periods of time [8]. The accumulation of hemozoin in the circulation has been shown to correlate well with disease severity and also its short and long term dynamics within leukocytes have been studied to some detail, yet not exhaustively [10,11,12]. Ultimately the crystals, encapsulated in tissue macrophages, are deposited in the spleen, liver and bone marrow.

Accordingly, the hemozoin production and clearance rates determine the optically measurable hemozoin concentration in the blood stream at any given time point. Our former in vitro measurements showed a significant correlation between the magneto-optical signal and parasitemia in case of synchronized *P. falciparum* cultures [3], but the question arises whether this correlation is preserved in the circulation in vivo. Therefore, it is crucial to evaluate the RMOD data relative to the parasitemia levels in murine parasites that have a synchronous blood stage cycle.

In this study, the relation of the RMOD signal to the parasitemia values of the self-limiting synchronous blood stages of *Plasmodium vinckei* lentum strain were compared to the parasitemia progression of the non-lethal asynchronous *Plasmodium yoelii* 17X-NL strain and the lethal synchronous *P. berghei* ANKA. The sensitivity of the RMOD technique was also tested during the treatment of *P. vinckei* infections with chloroquine, a compound which interferes with the formation of hemozoin [13]. Therefore, for the first time, this study evaluates the use of the RMOD technique in vivo to characterize the hemozoin formation during parasitemia progression and clearance following antimalarial treatment for malaria parasite models that have a synchronous blood stage cycle.

## 2. Materials and Methods

### 2.1. Ethics Statement

All animal care and protocols were conducted according to the approved protocols of the Institutional Animal Care and Use Committee (IACUC) of Tulane University. All other experimental protocols were performed according to the approved protocols of the Institutional Biosafety Committee (IBC) of Tulane University.

### 2.2. Mice and Parasites

Female BALB/c and Swiss Webster mice (six to eight weeks old) were purchased from Envigo Research Models and Services, formerly known as Harlan Laboratories, (Indianapolis, IN, USA). The non-lethal *P. vinckei* lentum strain and the *Pyp230p(–)* GFP expressing *P. yoelii* 17X-NL strain were stored as frozen stocks in liquid nitrogen [14,15]. Freshly thawed parasites were injected into donor naive mice, and a few days later 10,000 infected red blood cells (iRBCs) from donor mice were intravenously injected into the recipient experimental mice [16].

To establish the first time point of parasite detection, blood was collected from tail vein of the recipient experimental mice in 6–12 h intervals in the first 36 or 40 h [17]. Afterwards mice were monitored every 24 h or every 8 h for experiments with *P. vinckei,* the latter reported in Section 2.3. For the drug treatment experiments, chloroquine treatment started after 112 h post infection by providing chloroquine diphosphate salt (Sigma, St. Louis, MO, USA) in drinking water, at a concentration of 300 mg/L, with supplementation of 1% sucrose and with a pH adjusted to 3.5, for the whole treatment duration.

All *P. berghei* experiments were conducted in the rodent facilities of the Instituto de Medicina Molecular, Faculty of Medicine, University of Lisbon and were approved by the Ethical Committee of the Faculty of Medicine, University of Lisbon, as previously published [4].

### 2.3. Parasitemia Determination by Light Microscopy

At each blood sampling, parasitemia was estimated by counting 30 fields; i.e., approximately 9000–12,000 RBCs in Giemsa-stained thin blood were filmed by two microscopists independently [18,19]. The percentage of parasitemia was calculated according to the standard definition and the values obtained by the two counts were averaged.

### 2.4. Parasitemia Determination by Flow Cytometry

At each time point, 1 μL of blood was collected from the tail vein of mice infected with GFP-expressing *P. yoelii p230p(–)* for flow cytometric analysis. Blood samples were 1:2000 diluted in Roswell Park Memorial Institute (RPMI) medium and the percentage of green fluorescent erythrocytes were determined by flow cytometry. The green fluorescence protein (GFP) excitation was excited at a wavelength of 488 nm; however, the fluorescence was detected using a filter of 530 nm with the photomultiplier tube voltage set to its maximum sensitivity.

### 2.5. Parasitemia Determination by Rotating Magneto-Optical Measurements

For the magneto-optical measurements, 30 μL of tail vein blood was transferred from each mouse directly into a tube containing 2 μL of 1× heparin sodium salt (Sigma). Then 20 μL of blood was 20× diluted in the lysis solution containing 0.066 *v*/*v*% Triton X-100 in 11 mM NaOH. The RMOD measurements were performed using a volume of approximately 350 μL of each lysed sample. The detection limit was determined as the mean plus two times the standard deviation (95%) of all the RMOD signals measured on uninfected control mice of all the rodent experiments, performed using the RMOD method (RMOD values of approximately 20 independent uninfected blood samples). The average of the uninfected values and the corresponding detection limit are plotted in all graphs as black lines and dashed black lines, respectively.

The scheme of the RMOD setup, as well as the underlying physical principles of the detection, are described in detail in our former studies [2,3]. Briefly, the lysed sample, filled into a cylindrical sample holder, is inserted into the center of a ring-shaped assembly of permanent magnets, which creates a strong uniform magnetic field (B = 1 T) at the sample position. This magnetic field induces the co-alignment of the hemozoin crystals and when the magnetic ring is rotated, the co-aligned hemozoin crystals follow this rotation. During the measurement, polarized light from a laser diode is transmitted through the sample in the direction perpendicular to the plane of rotation of the magnetic field. The rotation of the co-aligned dichroic crystals gives rise to a periodic change in the transmitted intensity, which corresponds to the measured RMOD signal.

## 3. Results

### 3.1. Rotating Crystal Magneto-Optical Diagnostic Signal Based on the Hemozoin Production Is Proportional to the Parasitemia Percentage

In order to investigate the effect of transient hemozoin clearance on the relationship between parasitemia and the RMOD signal in vivo, the hemozoin production was monitored during the blood-stage infection of the non-lethal, synchronized *P. vinckei* lentum and the non-lethal, unsynchronized *P. yoelii* 17XNL strain. The results obtained were compared to our former results, obtained on the lethal strain *P. berghei* ANKA [4].

Figure 1A displays the RMOD signal versus the parasitemia for all individual mice investigated at various time points in the different experimental series. The RMOD signal, which is a direct measure of the hemozoin content of the blood samples, shows an overall proportionality with the parasitemia at any given parasitemia level above 0.01%; i.e., the sensitivity threshold of light microscopy. Since the parasitemia values below the aforementioned 0.01% were not possible to define, we cannot draw any conclusions about the performance of the RMOD method in this very low parasitemia region.

The relatively broad distribution/spread of the RMOD values corresponding to a given parasitemia level likely originates from the substantial difference between the hemozoin contents of parasites of different stages, belonging to the same parasite life cycle. There is a stepwise increase in parasitemia at the beginning of each cycle, typically by a factor of eight-to-ten [20]. However, the hemozoin production rate increases throughout the life cycle, being most pronounced at the trophozoite stage. Therefore, without specifying the stage of the parasites; the relationship between the RMOD signal and the parasitemia has approximately one order of magnitude of ambiguity, which is the difference of the hemozoin content at the beginning and at the end of the given cycle for a given parasite population.

Considering the high variability of parasite quantification by light microscopy, and the inherent difference between the two methodologies described above, the good correlation of all the measured RMOD and parasitemia values (Pearson’s correlation coefficient r = 0.8743, *p* < 0.05) indicates that the hemozoin content of the blood sample is primarily determined by the parasitemia in vivo. This observation implies that the quantitative interpretation of RMOD data is possible in the case of living organisms, similar to our previous findings on cell cultures and synthetic β-hematin crystals [1,2]. 

The characteristic time of hemozoin clearance after treatment, when viable parasites are not observable in the bloodstream anymore, is approximately 48–72 h [4,10,11,21]. Accordingly, during the progression of the infection a substantial amount of extra-erythrocytic hemozoin is always present in the bloodstream.

The linear dependence of the RMOD values on the parasitemia is well discernible over four orders of magnitude of the parasitemia in the double logarithmic plot of Figure 1A. This is the clearest at later stages of the infections when the parasite age distribution broadens within the peripheral circulation, suggesting that there is a dynamic equilibrium of the immune system’s clearing mechanisms and the hemozoin production rate of the parasites; i.e. there is no long-term accumulation of the hemozoin in the blood stream. The clearance rate of the plasma’s hemozoin content is roughly proportional to the actual parasitemia level, thus to the amount released during merozoite egresses the preceding erythrocytic cycle.

Besides the common tendency that the RMOD signal grows linearly with the parasitemia, the RMOD values corresponding to the four different series show slight systematic differences, which may indicate different hemozoin production rates for the different species. However, this difference is also significant for the two *P. berghei* ANKA series. Therefore, the observed differences are more likely to be attributable to a divergence in the age distribution of the peripheral parasites in the particular series, rather than a definite difference of the hemozoin production rates of the three investigated parasite species. In summary, there was no detectable difference in the average hemozoin production rates of the investigated three species.

The RMOD data in Figure 1A are replotted in Figure 1B,C as a function of time during the progression of infections. In the *P. yoelii* experiments, the first positive RMOD signal was detected at 30 hours. Since the length of one asexual cycle of *P. yoelii* is approximately 18 h [22], the RMOD method was able to detect the onset of the infections in the second erythrocytic cycle, after one multiplication event.

The first positive RMOD signal that unambiguously exceeded the detection limit (dashed black line) in *P. vinckei* infected mice was measured at 65 h. In the case of *P. vinckei,* the average length of one asexual cycle is approximately 24 h [22]. Therefore, the first positive signal detected by the RMOD device for *P. vinckei* belongs to the third erythrocytic cycle, following two multiplication events. The average number of merozoites emerging from a *P. vinckei* schizont is approximately one-third of the amount emerging from a *P. yoelii* schizont [23]. This difference in the average multiplication rate is also reflected in the less steep increase of the RMOD signal-versus-time curves measured for *P. vinckei* infections than those measured in the case of *P. yoelii* (black and grey data series in Figure 1B, respectively), and is probably responsible for the delayed detection of the onset of the blood stage in the *P. vinckei* infections compared to the other two species.

In case of the *P. berghei* infections, the time points of the first positive detections depend on the experimental settings [4,24,25]. In the first two experimental series, when the infections were carried out by the injection of 50,000 sporozoites, the first wave of erythrocytic parasites was already observed around 60 h post infection, which corresponds to 18 h after merozoites egress, since the liver stage is approximately 48 h long [26].

For *P. berghei* and *P. yoelii* blood stage infections’ time dependence of the RMOD signals (the average slope of the grey and the black curves in the semi-logarithmic plots of Figure 1B,C, respectively) is very similar, which indicates that the total hemozoin concentration of their circulation changes exponentially with a similar time constant over time. Since the average inter-cyclic multiplication rates for these species are alike, the identical slopes of the RMOD signal-versus-time curves in the semi-logarithmic plots imply that they are also very similar in terms of hemozoin production and clearance rates.

### 3.2. Monitoring Self-Resolving and Chloroquine-Treated Blood Stage Infections

Besides detecting the early stages of the infection, the correlation between the RMOD signal and the parasitemia was investigated for self-resolving rodent malaria strains for longer time periods as well. In these experiments P. vinckei lentum and GFP expressing P. yoelii 17X-NL infections were monitored by the RMOD method, light microscopy and flow cytometry (Figure 2 and Figure 3) for several days with daily blood samplings after the tail-vein injection of parasitized red blood cellss [27].

In case of the *P. yoelii* 17X-NL infection, presented in Figure 2, the time evolution of the RMOD signal over a long period again closely followed that of the parasitemia (r = 0.82, *p* < 0.05). An intensely progressing period of the infection is indicated by both the RMOD and the parasitemia values between days 1 and 5. After day 5, stagnation occurs both in hemozoin production and parasite multiplication, which is followed again by a period of moderate increase. A parasitemia level of 21% was reached at day 9.

The self-resolving *P. yoelii* 17X-NL infections are characterized by parasitemia curves consisting of sections of rapid growth, followed by a stagnation, and another growth, followed by saturation and subsequent decrease, as the infection gets cleared by the immune system [28,29,30]. The periods determined by the RMOD method; namely, an intense growth followed by a stagnation and recurrent increase are in agreement with literature data, showing approximately the first, progressing half of the complete course of the infection.

During the investigation of the *P. vinckei* lentum cases, displayed in Figure 3A, the first two thirds of the complete course of the approximately 20 day long self-resolving infection were monitored. After the onset of the infection, the RMOD signal and the parasitemia increased in parallel until day 8. The expected decline both in the parasite burden and hemozoin production was observed upon reaching the ~35% maximum parasitemia on day 9. After day 9, a drastic decrease is visible in the parasitemia values, which are closely reflected by the RMOD signals as well (overall correlation of the two datasets is r = 0.89, *p* < 0.05). *P. vinckei* infections usually reach higher peak parasitemia values compared to *P. yoelii* infections, which is supported by our results, and might be explained by the fact that *P. yoelii* parasites prefer to infect reticulocytes, while *P. vinckei* parasites rather infect mature red blood cells [29,30,31].

In an independent experimental series, the onset of the blood stage infection and the course of treatment were investigated by the RMOD method and microscopy (Figure 3B) in *P. vinckei* infections. After the injection of mice with *P. vinckei* lentum parasitized red blood cells, the development of the blood stage infection was followed for 112 h post-infection. After the sampling at 112 h, chloroquine treatment was applied in order to examine the hemozoin clearance that accompanies parasite elimination in treated mice. The first effects of drug treatment are indicated by the stagnation of the parasitemia value in the next 120 h time point, while a much stronger impact on the average hemozoin curve can be observed at the same time. The rapid decrease of the RMOD signal in the early hours of the treatment implies that the drug effect is already detectable after 24 h as expected in the case of chloroquine sensitive strains. Furthermore, its action can be revealed more directly by RMOD than by light microscopy, as the recession of the infection is not yet reflected in the smears at 136 h, contrary to the RMOD signal. This significant decline in hemozoin concentration is steadily observable until 184 h, and simultaneously a very steep decrease in the parasitemia is confirmed after the 136 h time point. Indeed, from 184 h post-infection, parasites were completely absent based on the microscopy results. The RMOD signal, however, is still positive at 184 h, implying that hemozoin is present in the circulation for at least one day after parasite-elimination. The overall characteristics of the RMOD signal and microscopy curves in this experiment further support the existence of a dynamic equilibrium between the effect of the immune system’s clearing mechanisms and the hemozoin production rate of the parasites, and that hemozoin content is mainly, but not solely determined by parasitemia.

In summary, the significant agreement between the parasitemia percentages and the RMOD curves in all three experiment series implies that the later might prove as an easy-to use, rapid and cheap alternative for monitoring the progression of in vivo malaria parasite infections.

### 3.3. Tracing Transient Hemozoin Clearance during the Intra-Erythrocytic Stages with High Temporal Resolution

The asexual developmental stages, such as pre-erythrocytic and intra-erythrocytic cycles occur in the mammalian host with exact periodicity. The asexual blood stage cycle starts with the invasion of the RBCs by merozoites released from hepatocytes. After the invasion of red blood cells, their maturation results in ring, trophozoite and schizont stages, respectively. At the end of the cycle merozoites rupture the mature schizonts and the host RBCs, and invade naive RBCs to start a new cycle [32].

To determine whether the RMOD technique can detect quite low changes in hemozoin concentration within one intra-erythrocytic cycle, the synchronous blood stage infection of the *P. vinckei* lentum strain was analyzed in an experimental series independent from the one presented in Section 2.1 and Section 2.2. In this new infection experiment presented in Figure 4A, 10,000 parasitized RBCs, with parasites in the trophozoite developmental phase, were injected into the mice.

By the examination of the smears at 42 h post-infection, early trophozoites were detected by light microscopy. Following the curve of the parasitemia after this time point, a step-like growth is observable between 48 and 60, 72 and 84 and between 96 and 108 h, respectively. Considering that the transformation of the parasites between the different developmental phases occurs every 7–8 h within the 24-hour blood stage cycle, the parasites reached the early ring stage with little or no freshly produced hemozoin in the beginning of the above listed timeframes (i.e., around 48, 72 and 84 h).

In parallel with the step-like growth of the parasitemia curve, the RMOD signal shows local minima, but in a delayed fashion (at 60, 84 and 108 h). This is caused by the periodic features of parasite development observable in synchronized infections, such as *P. vinckei*. Within one erythrocytic cycle parasites produce increasing amounts of hemozoin, as clearly reflected by the increasing RMOD signal. The parasites contain the most hemozoin at the schizont stage, and as a synchronized strain, most parasites reach this stage simultaneously. After the release of merozoites, the parasitemia rapidly increases; however, parasites in the early ring stage do not produce detectable amounts of hemozoin for approximately 6 h. On the other hand, simultaneously with merozoite egress, significant amounts of hemozoin crystals are released into the plasma during the rupture of the infected RBCs that are partially phagocytosed and gradually removed from the bloodstream. This clearing effect is manifested in slight drops (local minima) of the RMOD signal, observed at 60, 84, and 108 h. Similar stepwise changes of the RMOD signal observed in a previous study on *P. berghei* parasites are also shown in Figure 4 for comparison. *P. berghei* infections are considered to be asynchronous, especially in advanced stages; however, when we infected the mice with sporozoites and followed the changes of hemozoin concentration in the blood stream by the RMOD method within the first two erythrocytic cycles, the slight drops in the signals at 72 h and 90 h post-infection indicate that a partial synchronicity is likely to be preserved in the beginning of the blood stage, similarly to the *P. vinckei* infections.

## 4. Discussion

Several techniques are being developed for the reliable and rapid diagnosis of malaria [33,34,35,36] by exploiting the specific magnetic and optical properties of hemozoin crystals produced by all *Plasmodium* species during their asexual replication in the blood. In our previous studies, the performance of the rotating crystal magneto-optical method was tested with synchronized *P. falciparum* in vitro cultures and asynchronous *P. berghei* in vivo mouse infections [2,3,4] in order to assess the technique’s potential as a laboratory tool and diagnostic technique for malaria detection. In this study, we have extended our work to further rodent *Plasmodia*, which resembles the synchronized nature of the blood stage developmental characteristic for human infections [37].

Hemozoin, in addition to being considered as a biomarker for diagnostic tools, was also found to be a biologically efficient immune modulator [38]. The application of hemozoin as an indicator of self–limiting parasitic infection was elucidated here by following the progression of the self-limiting infections of non-lethal asynchronous *P. yoelii* 17X-NL strain, the synchronous *P. vinckei* lentum strain and the lethal asynchronous *P. berghei* ANKA strain. The time-dependent RMOD curves reflected the changes in parasitemia very closely, both in the progressing and the declining periods of the blood stage. Furthermore, the action of chloroquine during the treatment of *P. vinckei* infections could be revealed more directly by RMOD than by light microscopy, as the recession of the infection was detected roughly one period sooner via the former technique. The RMOD signal’s short response time suggests that the hemozoin crystals are released from the dying parasites quite rapidly and they are filtered out from the bloodstream in detectable quantities within a single cycle.

While monitoring the early stages of synchronous parasite development in *P. vinckei* infected mice, the RMOD signal could track the exponentially increasing hemozoin production of the parasites within a given cycle and indicate transient changes caused by subsequent clearance of free hemozoin or hemozoin containing phagocytes after merozoite egress.

Besides the species specific, time-dependent monitoring of the infections, the pooled RMOD values of all strains were found to show good a correlation with the parasitemia (r = 0.8743, *p* < 0.05). Furthermore, the different multiplication rates of *P. vinckei* and *P. yoelii* strains were also observable in the RMOD curves recorded over multiple days of the infection. The broadening of the RMOD versus parasitemia point distribution can be readily understood by considering that the amount of the hemozoin produced within a cycle may increase even by one order of magnitude, while the parasitemia only changes at the end of the cycle in a synchronous infection.

In summary, the progression of in vivo murine infections can be monitored by the RMOD method with high sensitivity, and its capability of determining drug efficacy was shown to be comparable to that of light microscopy. Thus, having further advantages such as being rapid and automated, its application as a diagnostic method and drug testing tool might be worth considering both in laboratory experiments and in-field conditions. These results further support the applicability of the RMOD method as a tool for the study of *Plasmodium* in vivo infections, holding out a promise for in-field diagnosis of human infections. 

## Figures and Tables

**Figure 1 biomolecules-09-00579-f001:**
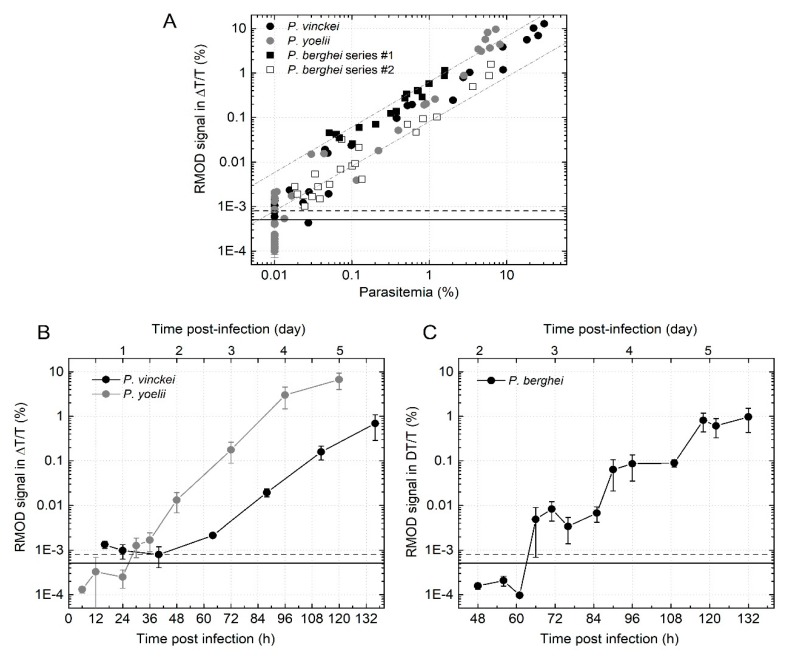
Analysis of the relationship of parasitemia and rotating-crystal magneto-optical diagnostic (RMOD) signal and the hemozoin production rates of three rodent *Plasmodium* species. (**A**) The RMOD signals plotted as a function of parasitemia. Each point shows a measurement on one mouse of a given series in a given time point. Black and gray circles represent *Plasmodium vinckei* and *P. yoelii* infections, and empty and filled black squares display two independent *P. berghei* infection experiments, respectively. Data on *P. berghei* are reproduced for the purpose of comparison from reference [4]. (**B**) The progression of the blood-stage infections of *P. vinckei* and *P. yoelii* parasites in vivo, monitored by the hemozoin production over time. Each circle and error bar represents the average and standard deviation of values measured on blood samples from *n* = 3 and *n* = 4 mice for the *P. vinckei* and *P. yoelii* series, respectively. (**C**) The progression of the blood-stage infections monitored by the hemozoin production of *P. berghei* parasites in vivo. Each circle and error bar represents the average and standard deviation of values measured on blood samples from *n* = 4 mice. (reproduced from reference [4]). In all graphs the continuous black line represents the average of the RMOD values of ~20 uninfected control samples. The dashed black lines indicate the detection limit of the RMOD method defined as the average plus two times the standard deviation of the uninfected values. The parasitemia values were determined either by microscopy or by flow cytometry.

**Figure 2 biomolecules-09-00579-f002:**
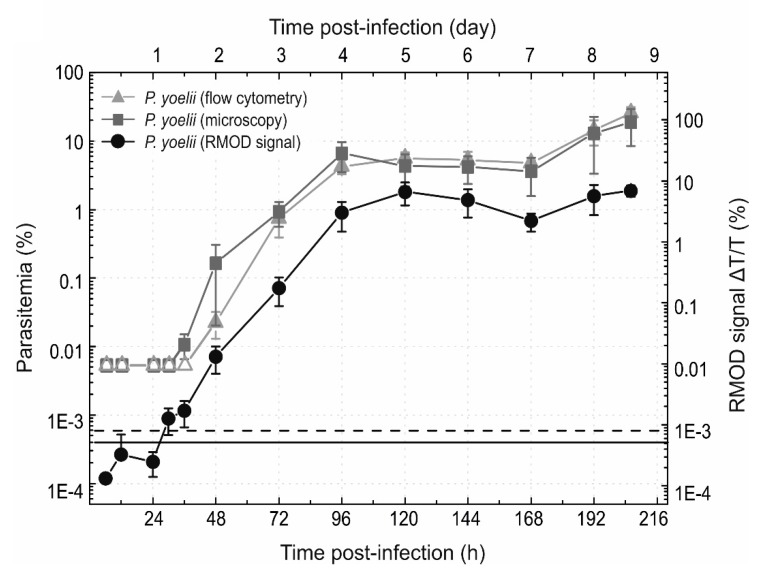
Progression of the blood-stage infection in mice injected with *P. yoelii* parasitized RBCs. The triangles represent parasitemia values measured by GFP-based flow cytometric detection; the squares represent parasitemia values determined by microscopy and the circles display the RMOD signal. The empty squares and triangles show samples declared negative either by microscopy after counting 10,000 red blood cells, or by flow cytometry. Each symbol and error bar represents the average and standard deviation of values measured on blood samples from *n* = 4 mice. The continuous black line represents the average of the RMOD values of ~20 uninfected control samples. The dashed black lines indicate the detection limit of RMOD defined as the average plus two times the standard deviation of the uninfected values.

**Figure 3 biomolecules-09-00579-f003:**
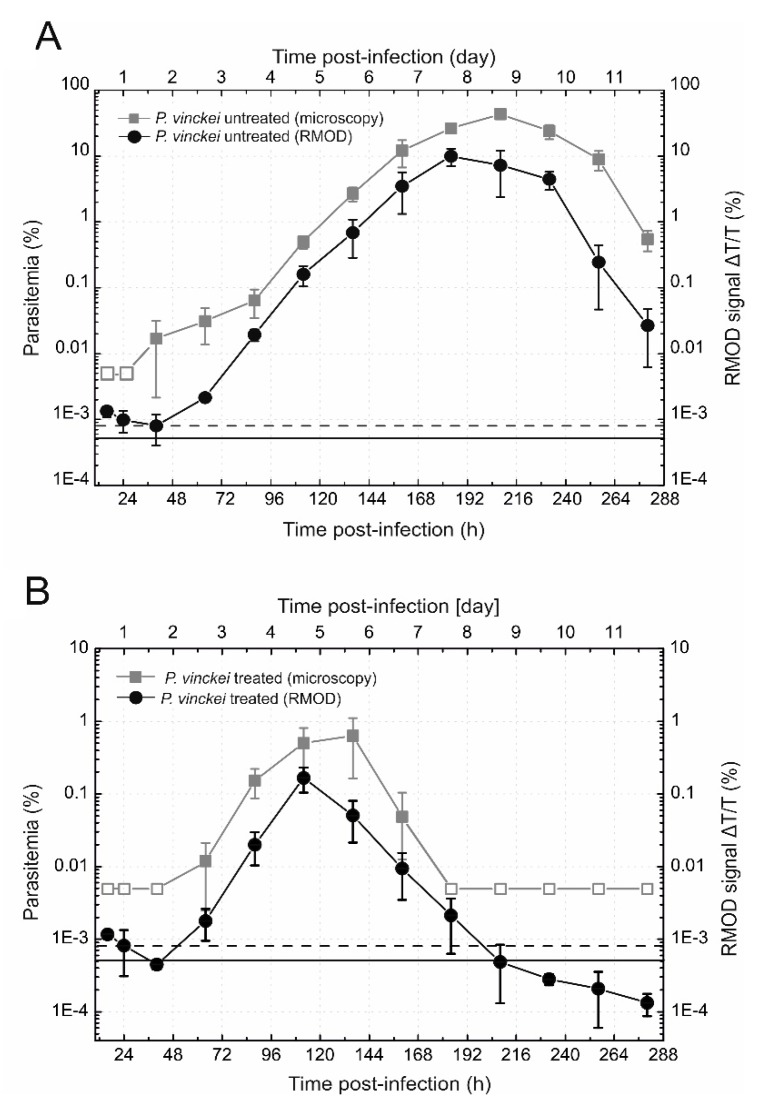
Monitoring of the self-resolving *P. vinckei* blood-stage infection and its course during chloroquine treatment. (**A**) Monitoring the development of the *P. vinckei* infection. The gray squares represent the parasitemia values determined by microscopy; the empty squares show samples declared negative after counting 10,000 red blood cells. The black circles represent the RMOD signal. (**B**) Monitoring the effect of chloroquine treatment in *P. vinckei* infections. The gray squares represent the parasitemia values determined by microscopy; the empty squares show the negative samples. The black circles represent the corresponding RMOD signal. The chloroquine treatment started after the sampling at 112 h post-infection. Both in panel (**A**) and (**B**), each symbol and error bar represents the average and standard deviation of values measured on blood samples from *n* = 4 mice. The continuous black line represents the average of the RMOD values of ~20 uninfected control samples. The dashed black lines indicate the detection limit defined as the average plus two times the standard deviation of the uninfected values.

**Figure 4 biomolecules-09-00579-f004:**
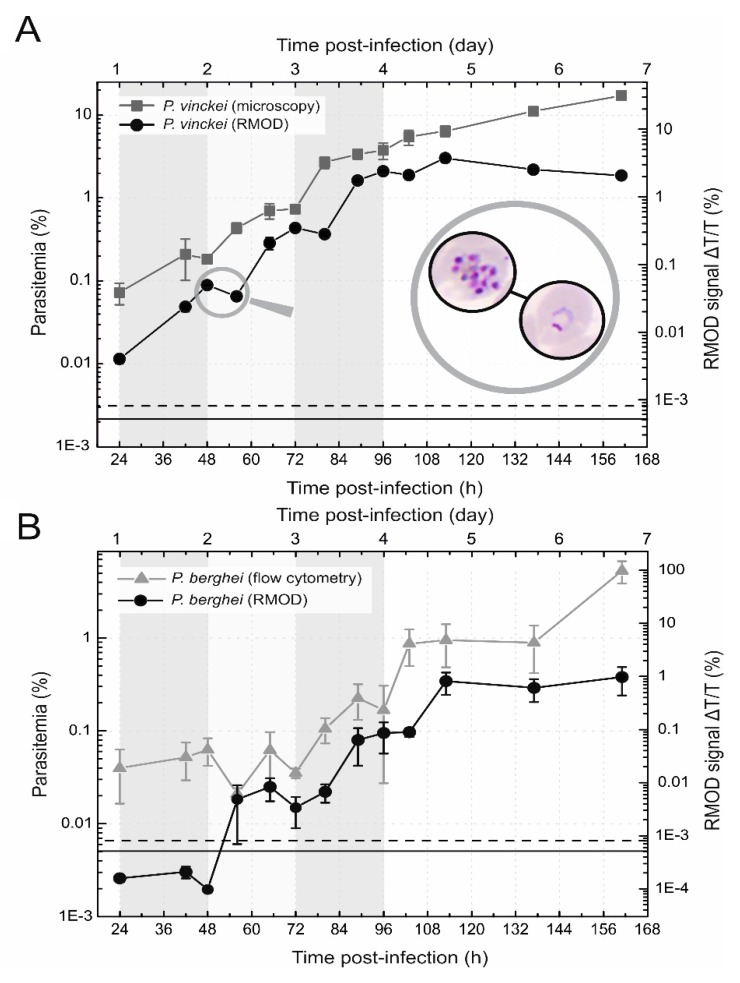
Detection of cyclic development of intra-erythrocytic *P. vinckei* and *P. berghei* infection. (**A**) The cyclic development of the *P. vinckei* infection. The black circles represent the RMOD signal, while the gray squares represent the parasitemia values determined by microscopy. Each symbol and error bar represents the average and standard deviation of values measured on blood samples from *n* = 5 mice. The magnifying glass shows the pictures taken during the examination of smears at 48 h and at 56 h post-infection, displaying the parasitic stage most representative of the given smear; i.e. schizonts and rings, respectively. (**B**) The cyclic development of *P. berghei* parasites at the beginning of the blood stage infection. The black circles represent the RMOD signal, while the gray triangles represent the parasitemia values determined by flow cytometry. Each symbol and error bar represents the average and standard deviation of values measured on blood samples from *n* = 4 mice. The background shading indicates the assessed layout of the first three 24-hour erythrocytic cycles. The continuous black line represents the average of the RMOD values of ~20 uninfected control samples. The dashed black lines indicate the detection limit defined as the average plus two times the standard deviation of the uninfected values.

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
