# Peer review of "Highly Sensitive and Rapid Characterization of the Development of Synchronized Blood Stage Malaria Parasites Via Magneto-Optical Hemozoin Quantification"

_biomolecules, 2019, doi:10.3390/biom9100579_

Round 1

Reviewer 1 Report

This manuscript reports on the application of a rotating-crystal magneto-optical diagnostic (RMOD) method to in vivo parasitemia measurements in mouse models of malaria. The authors have previously developed and reported on the RMOD method, which measures hemozoin content and shown it to be capable of measuring parasitemia both in P. falciparum cultures and in blood infected with P. berghei, a mouse malaria that features unsynchronized parasites. In the present study, the authors have used RMOD to measure hemozoin content in the blood of mice infected with P. vinckei and P. yoelii. The former resembles human malaria in so far is the parasite grows synchronously in the host. The authors have taken advantage of this to explore how the RMOD signal varies with parasitemia and time post-infection as well as in response to the antimalarial chloroquine. This is a necessary step towards eventually validating this method with human blood.

In this study, the authors have directly compared the performance of the RMOD technique with two gold-standard methods for measuring parasitemia, namely light microscopy of Giemsa stains and flow cytometry (in instances where this could be used). In all cases, they convincingly show very similar responses in RMOD and microscopy/flow cytometry. The study is convincing in showing that hemozoin in these mouse malaria species is directly related to parasitemia and can be used to measure parasitemia. Pleasingly, the authors have not made claims that this would automatically translate into success in humans. Rather, they have been careful to point out that the method would be useful in laboratory settings where mouse malaria is used as a model. The findings are of course promising for malaria diagnosis in humans and the authors indicate that further studies are warranted because of this. I would agree with this statement. On the other hand, sequestration of P. falciparum trophozoites and schizonts in humans, may substantially undermine the relationship between hemozoin levels and parasitemia in human blood. For this reason, the cautious approach adopted by the authors is warranted and to be welcomed.

Given the need for improved malaria diagnostics, with respect to ease of use and sensitivity, this study is to be welcomed and is likely to be of considerable interest.

There are a few minor corrections needed as follows:

Lines 143, 148, 162, 169: the word “strains” should be “species”

Line 186: “literary data” should be “literature data”

Lines 222, 232, 263, 285: I would suggest rather using “post-infection” than the abbreviation “pi” which has not been defined in the text.

Line 326: “Materilas” should be changed to “Materials”

Reviewer 2 Report

The manuscript entitled “Highly sensitive and rapid characterization of the development of synchronized blood stage malaria parasites via magneto-optical hemozoin quantification” describes the rotating-crystal magneto-optical diagnostic (RMOD) technique used as a quantitative tool to follow the developmental progress of Plasmodium malaria parasites. The manuscript is written well and needs a few minor edits to clear up experimental work. However, the findings are sound and the science is well explained. The novelty of the work however is the largest issue.

The background information that lead to the work are explained well. It also clearly states that this is a follow-up publication where previous work has been done using this technique to evaluate parasitemia in synchronized Plasmodium falciparum cultures. The introduction states that this study is the first time that the RMOD technique is used to characterize hemozoin formation in synchronous parasites in vivo. This included looking into parasite progression and clearance following antimalarial treatment for malaria models that have a synchronous blood stage cycle.

A previous publication of this work from the same authors in 2016 (Sci Rep. 2016 Mar 17;6:23218. doi: 10.1038/srep23218), described the use of RMOD to evaluate in vitro P. falciparum cultures. The RMOD technique was then further used to monitor in vivo onset and progression of the blood-stage infection in Plasmodium berghei. In addition, the study in 2016 also looked at the treatment of P. berghei infections and how the RMOD technique can be used for the detection of parasites after treatment. Although it is clearly stated in the submitted manuscript currently under evaluation that the authors are now investigating using the RMOD in three more cultures strains (i.e. unsynchronized non-lethal Plasmodium yoelli 17X-NL, synchronise Plasmodium vinkei and lethal synchronise Plasmodium berghei ANKA) that is more synchronise in order to evaluate the technique to resemble human malaria infections like P. falciparum and Plasmodium vivax, the experiments done resemble those published in 2016 in that it again looked at onset and progression as well as the evaluation of chloroquine treatment. This raises concerns about the novelty of the work presented here as the only difference is using more synchronous parasite lines. If these experiments where included in the 2016 publication it would have made for a much stronger body of work than separating it into two publications. This publication seems to be an extension of similar findings (as stated in discussion, page 10, line 294) than before.

To address this shortcoming, this reviewer would like to suggest that the authors need to re-evaluate how they can include novel experiments to show the value of this tool for in-field diagnosis of human infections and to illustrate the applicability of the RMOD method in the field. The technique clearly has potential, but what would be the next step to implement this technique successfully in the field?

Other minor comments directly with the content of the manuscript are listed below:

Line 80…extra spaces between “…hemozoin” and “[30]”

Figure 1. The resolution of this figure seem to be low.

Figure 1. Blue squares are described in the figure legend, but                             no colour is visible.

Lines 119-122 The content in these lines seem as if it was intended for Figure 1 legend. It should be moved there.

Line 149-151   It is stated that for P. yoelii experiments the first positive RMOD signal was detected at 30 hours. However there are no 30  hour time points on the graphs.

Line 154-155   The first positive RMOD signal for P. vinckei was detected at 65 hours. However, from the labeling in Figure 1B it seems as if the first positive RMOD signal was detected for P. yoelliat 65 hours. This should be cleared up

Line 159-160   …”the average multiplication rate is reflected in the     less steep increase of the RMOD signal-versus-time                             curves..” If the rate is discussed a scientific value should be determined and using the phrase “less steep” should be removed. What does less steep mean? Less than what? What is it being compared to?

Line 168-170   What is being compared. If mulitiplication rates are being compared should you look at the total rate? What is being compared?

Line 174, 177  P. vinckei and P. yoelli in italics.

Line 187          “The significant agreement between…” Where stats done to determine how significant the agreement is?

Line 216          define pi when used for the first time

Line 236          font size changes towards the end of the line

Line 256          Should this read “parasite stage” instead of “parasite age” ?

Line 294         Give more support or justification on why the experiments done in this study and that in 2016 “assess the technique’s potential for in-field applicability”.

Line 312         Different font size for “pooled RMOD”

Line 314         See comment in for line 168-170. How can good correlation be seen if rates where not determined/calculated? How do you determine if the correlation is good or bad?

Line 319-325  Justify/explain/support why it is important to be able to determine drug efficacy with this technique rather than using light microscopy. How ill these findings translate to diagnostics in the field?

General comments:

All figures should include the amount of experiments done (n=?) and how the error bars for each data point was determined.

For all curves containing data points for RMOD signal, data points are shown below the dashed lines which indicates the detection limit of the RMOD technique. How can these values be indicated and an error be indicated if they fall below the detection limit? This should be clarified.

Round 2

Reviewer 2 Report

The authors have addressed the comments made in the first revision. I feel the publication can be accepted in present form.